The effects of venting and decompression on Yellow Tang (Zebrasoma flavescens) in the marine ornamental aquarium fish trade

Munday Emily S. 1 emily.munday@gmail.com
Tissot Brian N. 2
Heidel Jerry R. 3
Miller-Morgan Tim 3 4
1 School of the Environment, Washington State University Vancouver , Vancouver, WA , USA
2 Marine Laboratory, Humboldt State University , Trinidad, CA , USA
3 Veterinary Diagnostic Laboratory, College of Veterinary Medicine, Oregon State University , Corvallis, OR , USA
4 Aquatic Animal Health Program, Oregon Sea Grant/College of Veterinary Medicine, Oregon State University—Hatfield Marine Science Center , Newport, OR , USA
Toonen Robert
Electronic publication date: 2015 Feb 17
Publication date: 2015
Volume: 3
Electronic Location ID: e756
Received 2014 Nov 4; Accepted 2015 Jan 20
Copyright: © 2015 Munday et al.
Copyright year: 2015
Copyright holder: Munday et al.
License: This is an open access article distributed under the terms of the Creative Commons Attribution License, which permits unrestricted use, distribution, reproduction and adaptation in any medium and for any purpose provided that it is properly attributed. For attribution, the original author(s), title, publication source (PeerJ) and either DOI or URL of the article must be cited.
License URL: https://creativecommons.org/licenses/by/4.0/

Keywords: Aquarium fish, Venting, Fish physiology, Fish barotrauma, Fish cortisol, Aquarium trade

Funding: National Oceanic and Atmospheric Administration Coral Reef Conservation Program NA11NOS4820013 Washington State University Vancouver This research was supported by the National Oceanic and Atmospheric Administration Coral Reef Conservation Program [NA11NOS4820013] and Washington State University Vancouver. The funders had no role in study design, data collection and analysis, decision to publish, or preparation of the manuscript.

==============================
Each year, over 45 countries export 30 million fish from coral reefs as part of the global marine ornamental aquarium trade. This catch volume is partly influenced by collection methods that cause mortality. Barotrauma in fish resulting from forced ascent from depth can contribute to post-collection mortality. However, implementing decompression stops during ascent can prevent barotrauma. Conversely, venting (puncturing the swim bladder to release expanded internal gas) following ascent can mitigate some signs of barotrauma like positive buoyancy. Here, we evaluate how decompression and venting affect stress and mortality in the Yellow Tang (Zebrasoma flavescens). We examined the effects of three ascent treatments, each with decompression stops of varying frequency and duration, coupled with or without venting, on sublethal effects and mortality using histology and serum cortisol measurements. In fish subjected to ascent without decompression stops or venting, a mean post-collection mortality of 6.2% occurred within 24 h of capture. Common collection methods in the fishery, ascent without decompression stops coupled with venting, or one long decompression stop coupled with venting, resulted in no mortality. Histopathologic examination of heart, liver, head kidney, and swim bladder tissues in fish 0d and 21d post-collection revealed no significant barotrauma- or venting-related lesions in any treatment group. Ascent without decompression stops resulted in significantly higher serum cortisol than ascent with many stops, while venting alone did not affect cortisol. Future work should examine links in the supply chain following collection to determine if further handling and transport stressors affect survivorship and sublethal effects.

Introduction

Each year, over 45 countries remove and export 14–30 million fish from coral reefs as part of the marine ornamental aquarium trade (Bruckner, 2005; Wood, 2001). Although ∼90% of freshwater aquarium fish are successfully cultivated in aquaculture facilities, most tropical marine aquarium fish are wild-caught (Wood, 2001). Collecting live fish for the aquarium trade involves removing reef fish from SCUBA diving depths (∼10–35 m) to the surface. Collection is followed by transporting fish from the collection site to an export facility where they are held for 1–7 days prior to shipment. The fish are then packaged in plastic bags with enough water to turn around in and 100% oxygen, placed in boxes, and shipped to an import facility where they may be held for several days. The fish then are transported to a retail store and, finally, to a hobbyist aquarium. Mortality may occur at any point in this supply chain, impacting each participant in the industry, and negatively affecting coral reefs through increased collection pressure to replace losses (Stevenson, Tissot & Dierking, 2011; Tissot et al., 2010).

Aquarium fisheries that use destructive fishing practices (e.g. cyanide is used to stun ornamental fish for ease of capture) have high fish mortality, and this practice is still widespread (Hall & Bellwood, 1995; Hanawa et al., 1998; Rubec et al., 2001; Rubec & Cruz, 2005; Bell et al., 2009). While fishers in Hawaii do not use cyanide to collect fish (Walsh et al., 2004), and immediate mortality is low (<1%) (Stevenson, Tissot & Dierking, 2011), levels of delayed mortality are unknown. Because fish move rapidly through the supply chain, it is possible that aquarium fishers are unaware of collection methods that result in mortality further along the supply chain. Economically, delayed mortality shifts the burden of fish death and monetary loss from the collector to those further along the supply chain (e.g., the importer, or hobbyist) while also increasing the demand for fish and exacerbating pressure on coral reef ecosystems. Identifying methods that cause delayed mortality would reduce the overall mortality of aquarium fish in the aquarium trade, and thus the number of fish removed from the reef to compensate for these losses.

In order to identify industry methods that cause delayed mortality in aquarium fish, it is necessary to examine each link in the aquarium fish trade supply chain both independently and in succession. In this study, we examine the first step in the supply chain: removing fish from depth (15–18 m) to the surface. Mortality caused by removing live fish from coral reef depths to the surface is an important and controversial issue affecting the aquarium fishery; to our knowledge, ours is the first study to examine this problem.

To ensure that fish survive the transition from depth to the surface, aquarium fishers must either prevent or mitigate barotrauma. Fish experience barotrauma when they are brought to the surface. As water pressure decreases, the volume of swim bladder gas increases. This phenomenon is a result of Boyle’s Law, in which decreasing pressure causes an exponential increase in gas volume. Barotrauma signs in fish manifest both externally and internally, and include: positive buoyancy caused by overexpansion of the swim bladder; bulging of the eyes, or exophthalmia; and protrusion of the intestine from the cloaca. While barotrauma has not been studied in shallow-dwelling (15–18 m) reef fish caught for the aquarium trade, there is ample research on the effects of depth changes on deeper dwelling (20–152 m) fish caught commercially and recreationally for consumption (Table 1).

Table 1 External and internal signs of barotrauma observed in food fishes.

	External	Internal	
Gotshall, 1964	Esophageal eversion		
Bruesewitz, Coble & Copes, 1993	Esophageal eversion		
Keniry et al., 1996	Esophageal eversion, positive buoyancy		
St John & Seyers, 2005	Esophageal eversion, exophthalmia		
Parker et al., 2006	Esophageal eversion		
Hannah & Matteson, 2007	Esophageal eversion		
Hannah, Parker & Matteson, 2008	Esophageal eversion, exophthalmia		
Jarvis & Lowe, 2008	Subcutaneous gas bubbles, esophagealeversion, exophthalmia	Arterial embolism, hemorrhage, organ torsion	
Nichol & Chilton, 2006	Ruptured swim bladder		
Rogers et al., 2008	Exophthalmia	Damage to and displacement of organs surrounding swim bladder	
Pribyl et al., 2009	Esophageal eversion, exophthalmia	Emphysema of heart ventricle	
Wilde, 2009	Esophageal eversion		
Brown et al., 2010	Cloacal prolapse, exophthalmia, esophageal eversion		
Pribyl et al., 2011	Esophageal eversion	Emphysema of heart ventricle and epithelial surfaces, gas emboli in rete mirabile and head kidney	

Prior research has demonstrated that fish continue to exhibit sublethal injuries (not having caused death) for extended periods. Rupture of the outer layer of the swim bladder (tunica externa) persisted for at least one month after collection in rockfish (genus Sebastes) (Pribyl, 2010). This indicates that sublethal signs of barotrauma persist long after the initial trauma occurs. Knowing this, we predict that fish collected for the live ornamental aquarium trade also suffer sublethal injuries that could result in delayed mortality.

Because barotrauma can be potentially fatal to both shallower-dwelling aquarium fish and deeper-dwelling food fish alike, fishers implement methods that either prevent or mitigate it. Venting is a method that mitigates barotrauma and involves puncturing a fish swim bladder with a hypodermic needle to allow gases to escape the swim bladder, relieving positive buoyancy. Decompression, in contrast, is a method that prevents barotrauma. Decompression involves transporting fish from depth to the surface over a longer period of time, which allows expanding gases to be removed from the swim bladder, resulting in a fish that is not subjected to barotrauma at all. Fishers implement one or some combination of both of these methods in order to help fish survive the pressure transition. While the use of venting and decompression on aquarium fish has been documented (Randall, 1987; Pyle, 1993; LeGore, Hardin & Ter-Ghazaryan, 2005), ours is the first study to evaluate the efficacy of each of these procedures in preventing mortality.

In deeper-dwelling fishes, decompression takes a long time—up to several days (Parker et al., 2006; Pribyl, 2010). In order to prevent barotrauma, one must allow adequate time for fish to naturally remove gas from the expanding swim bladder. Likewise, in Hawaii where fishers collect shallow-dwelling reef fish, ascent with multiple decompression stops can be time-consuming (∼2 h). Providing fish time to naturally decompress and remove swim bladder gases prevents fishers from moving from one reef site to another because fish collection containers must be attached to the surface vessel. Fishers would rather remove fish from depth quickly in order to move to another reef site, return to depth, and collect more fish. However, bringing fish up to the surface quickly without decompression stops results in barotrauma. To mitigate this barotrauma, fishers use venting.

Research on deeper-dwelling food fishes disagree that venting reduces fish mortality. This is largely an artifact of the differences in species and depths the studies examine (Gotshall, 1964; Keniry et al., 1996; Shasteen & Sheehan, 1997; Collins et al., 1999; Kerr, 2001; Nguyen et al., 2009; Wilde, 2009) as well as differences in the length of time fish are observed in captivity. We predict that long-term holding will allow us to definitively conclude how collection methods affect fish health.

As previously stated, fishers often use some combination of decompression and venting. For example, it is common for aquarium fishers to perform one or several decompression stops, pausing in the water column at intermediate depths before removal to the surface (LeGore, Hardin & Ter-Ghazaryan, 2005; Stevenson, Tissot & Dierking, 2011). In Hawaii, fishers typically vent the fish following this practice. While these methods of barotrauma prevention and mitigation likely positively affect fish health and mortality, these practices are controversial among the animal rights community. Such groups in Hawaii have repeatedly proposed legislation that would ban the harvest of marine species for the aquarium trade based on animal cruelty claims (see Lauer, 2011; Talbot, 2012; Wintner, 2010; Wintner, 2011). Groups opposed to venting claim that it inflicts stress and mortality on fish, while collectors maintain that venting is necessary for fish survival. People who oppose venting have suggested that decompression be used instead. While we may not solve the values conflicts driving in this controversy, we do hope to inform pending management decisions related to aquarium fish collection in Hawaii.

In our study, we seek to: (1) Determine short- and long-term mortality of reef fish caught for the aquarium trade subjected to the barotrauma prevention and/or mitigation practices of decompression and venting, respectively; (2) Examine sublethal effects of collection that could result in delayed mortality.

Methods

Experimental design

This study was conducted on the west coast of the island of Hawaii in June–July 2011. The Yellow Tang (Z. flavescens) was selected as the study animal because it is the most commonly targeted aquarium fish species in Hawaii, consistently composing nearly 80% of the total catch of aquarium fish there (Cesar et al., 2002; Tissot & Hallacher, 2003; Walsh et al., 2004; Williams et al., 2009). In addition, Acanthuridae, the family encompassing Yellow Tang and other surgeonfishes, is one of the most common families targeted globally in the live aquarium trade (Rhyne et al., 2012). Therefore, understanding how collection practices affect Yellow Tang health and survival is especially relevant to the marine aquarium fishery.

To examine the effects of collection practices on mortality of Yellow Tang, we used a fully crossed factorial experimental design. Three decompression treatments (no decompression stops, one decompression stop, many decompression stops) were coupled with venting (yes, no) in all possible combinations (k = 6 treatments) (Fig. 1). Each of the six treatments was replicated three times, with n = 20 fish in each replicate for a total of 360 individual fish.

Figure 1 Experimental design of ascent and venting treatments.

(A) Illustrates decompression stops for each ascent treatment. The rate of ascent between stops was 0.25 m/s for all treatments. Fish subjected to ascent without decompression stops were brought directly to the surface. Fish subjected to ascent with one decompression stop were brought to half the maximum depth for a 45 min decompression stop, then brought to the surface. Fish subjected to multiple decompression stops were brought up 3 m every 15 min. At 10 m (2 atm), these fish were brought up 1.5 m every 15 min. (B) Venting treatment scheme and fish sampling design. This experiment was replicated 3 times for a total sample size of n = 360 fish.

Yellow Tang were subjected to collection methods typical of the fishery, as elucidated through interviews with active aquarium fishers. Fish were collected between 15–18 m depth, reflecting the range frequented by Hawaiian collectors (Stevenson, Tissot & Dierking, 2011). In order to accurately reflect methods used by aquarium fishers, an aquarium fisher with over 15 years of experience collected the fish. One aquarium fisher performed the experiment in order to maintain consistent methods throughout the experiment. Working with several fishers would be ideal, but this was not logistically or financially feasible. Fish collection occurred on SCUBA using a barrier net, as described by Stevenson, Tissot & Dierking (2011). Following capture, fish were transferred to containers assigned to an ascent treatment. Following ascent to the surface vessel, half of the fish were vented and half were not.

Three ascent treatments were used: (1) ascent without decompression stops, (2) ascent with one decompression stop, and (3) ascent with multiple decompression stops. The rate of ascent between decompression stops was 0.25 m/s for all treatments, the recommended SCUBA ascent rate and the rate fishers ascend while transporting fish from depth to the surface. Fish subjected to ascent without decompression were brought directly to the surface from depth. Fish subjected to ascent with one decompression stop were brought up to half the maximum depth for a 45 min decompression stop, and then brought to the surface. Fish subjected to multiple decompression stops were brought up 3 m every 15 min. At 10 m (2 atm), these fish were brought up 1.5 m every 15 min because the volumetric change resulting from the decrease in pressure is especially great the last few meters of ascent. As is typical in the fishery, venting was performed by the fisher on the fishing vessel using a 20 G hypodermic needle, which was replaced after approximately 50 fish. Each fish was held out of water for less than 3 s by the fisher while the needle was inserted through the body wall toward the swim bladder, caudal to the pectoral fin and ventral of the lateral line. Following collection, fish were transported in the vessel’s live well until they reached the holding facility. During collection and transit from collection site to port, fresh seawater was continuously circulated through the live well.

Initial histologic diagnostics were performed on fish (n = 5 in each replicate group) immediately upon arrival at the holding facility to determine baseline health as well as to assess the immediate effects of venting and decompression. All fish that died were examined histologically to identify lesions that could have contributed to death. A final histologic diagnostic examination was also performed on surviving fish at the conclusion of the holding period. Serum cortisol concentration was also measured upon arrival at the aquaculture facility. Serum cortisol serves as a proxy for stress in fish (Donaldson, 1981).

Holding Period

Post-collection, fish were observed for 21 d at an aquaculture facility located at the Natural Energy Laboratory Hawaii Authority (NELHA) in Kona, Hawaii provided with natural surface seawater at ambient temperatures. The experimental duration was chosen because after interviewing fishers operating in Kona, Hawaii, we determined that 21d represents a reasonable time period for a fish to be transferred from the reef to a retailer or hobbyist in this particular chain. In addition, swim bladder healing in rockfish has been observed after 21 d (Parker et al., 2006) and is sufficient time to allow skin and muscle regeneration in fish (Roberts, 2010). Therefore, fish exhibiting lesions after 21 d may not have fully recovered in a supply chain environment and could be categorized as having sublethal effects from collection.

Fish were held in 1 m diameter mesh floating cages within three 10,000 l pools, which served as replicate blocks, each containing all six treatments. Incoming seawater was filtered to 5 µm, and set to flow through each pool at a rate of 1 volume/d. Pools were exposed to natural sunlight, and temperature and salinity were measured twice daily. All fish were fed a natural algae diet (Ulva fasciata) rich in nutrients absorbed from food fish outflow in the aquaculture facility for biofiltration.

Fish were monitored daily and mortality was recorded. Standard length (SL) (from snout to base of caudal fin) of each fish was measured. Following mortality, fish were placed in 10% neutral buffered formalin for histopathology; the operculum was removed and body cavity opened to facilitate proper formalin fixation of the internal tissues. Moribund fish were humanely euthanized using an overdose solution (>250 mg/l) of tricaine methanesulfonate (MS-222) (Sigma-Aldrich, St. Louis, Missouri, USA).

Histopathology

To determine the sublethal effects of collection methods, fish (n = 5) were chosen randomly from each replicate treatment group immediately upon arrival to the holding facility (0 d) and at the end of the holding period (21 d) for histopathology. Fish used for histopathology were euthanized using an overdose solution of MS-222, placed on ice, and shipped within 48 h to Oregon State University’s (OSU) Veterinary Diagnostic Laboratory (VDL) for histologic examination. Fish that died during the experiment were fixed in 10% neutral buffer formalin as described above and examined.

Formalin-fixed fish were immersed for 24 h in Cal-Ex II (Fisher Scientific, Waltham, Massachusetts, USA) to decalcify bone, and serial cross sections were processed using standard histologic techniques, sectioned at 5 µm, and stained with hematoxylin and eosin. Brown-Hopps Gram stain was used as necessary to assess for the presence of bacteria. All slides were examined using a Nikon Eclipse 50i microscope (Nikon, Minato-ku, Tokyo, Japan). Histologic examination focused upon gill, heart, kidney, liver, swim bladder, and intestine.

Primary stress response

Because of the potential for cortisol concentrations to decrease when a stressor subsides, blood samples were collected from fish immediately upon arrival to the holding facility. Fish (n = 2) were anesthetized from each treatment replicate group using MS-222 prior to drawing 0.3–1.0 ml blood from the heart using a 25G 2.54 cm needle and 3 ml syringe. Cardiac puncture was necessary because the small size of the fish. Following blood sample collection, fish were euthanized using an overdose solution of MS-222. To determine Yellow Tang ocean baseline cortisol concentration, blood was collected from fish (n = 4) underwater on SCUBA at capture depth within 3 min of capture. Blood was injected into 3 ml vacutainer tubes with no additive (Becton-Dickinson, East Rutherford, New Jersey, USA), placed on ice, and centrifuged at 3,000 rpm for 10 min <1 h later. Serum supernatant was transferred to a clean vacutainer tube with no additive, placed on ice, and frozen <1 h later for ≤40 d in a non-frostless freezer, and transported overnight on dry ice to the OSU Department of Fisheries and Wildlife for analysis.

Serum cortisol concentrations were determined using radioimmunoassay (RIA) as described by Redding et al. (1984). Total binding, the ratio of the radiolabeled cortisol bound to the antibody to the total amount of radiolabeled cortisol in the sample, was 40%–50%. Samples showed adequate parallelism, and 3.9–500.0 ng/ml cortisol standards were used.

Statistical Methods

Statistical analyses were performed using the Minitab 15 Statistical Software program. To meet assumptions of normality and homogeneity of variance, data were transformed to square root (fish SL) or log (cortisol). A one-way t-test was used to compare mean cortisol concentrations of each treatment group with the ocean baseline parameter. A two-way ANOVA was used to compare mean cortisol concentrations, with decompression treatment and venting as fixed factors and replicate block as a random factor. Tukey’s multiple comparisons test was used to determine significant differences between levels within each factor.

Results

Mortality

Sizes of Yellow Tang  in this study ranged from 5.0–10.0 cm SL with a mean value of 7.2 cm (SE = 0.05 cm). Mortality occurred <24 h post-collection in fish subjected to ascent without decompression stops or venting, with a mean mortality of 6.2% (SE = 0.6%). No mortality occurred in the other experimental treatments.

The incidence of mortality was consistent with observations of the frequency and severity of external barotrauma signs. These included high frequency of positive buoyancy, bloating, prolapse of the intestine from the cloaca, and exophthalmia (Fig. 2) in fish subjected to ascent without decompression stops. Venting relieved positive buoyancy and vented fish became neutrally or negatively buoyant (Fig. 2).

Figure 2 Barotrauma signs observed in Yellow Tang following collection.

(A) Positive buoyancy before venting and neutral to negative buoyancy following venting (B) intestinal protrusion from the cloaca and (C) exophthalmia.

Histopathology

Histopathology of gill, heart, kidney, liver, swim bladder, and intestine failed to detect significant inflammation, necrosis, or gas embolism associated with barotrauma or venting in any treatment, in both the short- and long-term. A venting wound was detected in a fish subjected to ascent with many decompression stops and venting at day 0. However, this lesion consisted only of locally extensive necrosis of body wall musculature and a localized influx of neutrophils surrounding the needle track and not significant widespread infection (Fig. 3).

Figure 3 Histologic views of swim bladder tissues in vented Yellow Tang.

(A)–(B) Representative histologic views of normal Yellow Tang swim bladder tissues in vented fish at (A) 0 days and (B) 21 days. Note lack of any inflammation, edema, or necrosis in lateral body wall (bw), coelomic cavity (c) swim bladder (sb), and rete mirable (rm). (C) Histologic section of needle track in a Yellow Tang subjected to venting showing muscle cell necrosis, edema, and neutrophilic inflammation: (1) Needle track, (2) needle entry through coelomic cavity, (3) neutrophilic inflammatory response.

Primary stress response

The mean ocean baseline cortisol concentration was 8.9 ng/ml (SE = 4.96 ng/ml) and in some cases was at or below the detection limit for the assay (3.9 ng/ml). All treatment groups were significantly elevated above the baseline cortisol concentration (One-way t-test; p < 0.05). There was no significant interaction between decompression treatment and venting. Decompression treatment significantly affected cortisol concentration (Two-way ANOVA: F = 4.26; df = 2, 12; p = 0.03) (Fig. 4). Ascent without decompression stops resulted in a significantly higher mean cortisol concentration (M = 58.8 ng/ml, SE = 8.7 ng/ml) than ascent with many 15 min decompression stops (M = 35.5 ng/ml, SE = 5.3 ng/ml), with neither treatment being significantly different from ascent with one 45 min decompression stop (M = 35.2 ng/ml, SE = 4.3 ng/ml). Ascent without decompression stops produced the highest observed cortisol concentration (101.49 ng/ml), whereas the highest observed cortisol concentrations in fish subjected to one and many decompression stops were 59.09 and 68.03 ng/ml, respectively. While venting resulted in higher mean cortisol concentration (M = 47.7 ng/ml, SE = 6.9 ng/ml) than the no venting treatment (M = 38.2 ng/ml, SE = 4.3 ng/ml), this difference was not statistically significant.

Figure 4 Cortisol concentration (mean ± SE) by each treatment.

(A) venting and ascent treatments; (B) ascent treatments; and (C) venting treatments. Letter groups represent Tukey’s multiple range test results comparing means. All treatment groups were significantly elevated above the ocean baseline concentration of 8.9 ng/ml.

Discussion

With the objective of informing management on collection practices in the aquarium trade, our study focused on the short- and long-term mortality of reef fish subjected to decompression and venting as barotrauma prevention and mitigation practices, respectively. Overall, we found that venting prevented immediate mortality in fish subjected to ascent without decompression stops. Ascent significantly elevated serum cortisol above baseline concentrations, and ascent without decompression stops resulted in significantly higher serum cortisol concentrations than ascent with many stops. Venting, however, did not significantly affect cortisol concentration. In the following sections, we explain our results, suggest future research recommendations, and discuss implications for management of this fishery.

Mortality

We found that the methods commonly used in this fishery (ascent without decompression stops, or ascent with one decompression stop, followed by venting) resulted in no immediate or delayed mortality. Ascent without decompression stops followed by venting resulted in no mortality, while fish subjected to ascent without decompression stops and no venting was the only treatment group in which mortality occurred. Venting alleviated positive buoyancy in fish following ascent with no decompression stops and in this way mitigated barotrauma sufficiently to prevent short-term mortality. Neutral buoyancy allowed fish to control body position and avoid colliding with the transport container during transport from reef to harbor. This is in contrast to fish subjected to ascent without decompression or venting, which exhibited positive buoyancy and were at risk of acquiring secondary transport-related injuries.

Additional factors that may influence post-collection mortality, but are outside the scope of this study, include collection depth, body size, and species. We examined fish collected from 15–18 m depths, which is typical for the Hawaii Yellow Tang fishery, though fishers do exceed this range (i.e., ≥27 m) when targeting other species (Stevenson, Tissot & Dierking, 2011). At deeper depths, the effects of decompression and venting may differ, and it is known that fish mortality and occurrence of barotrauma increases with capture depth (Collins et al., 1999; St John & Seyers, 2005; Hannah, Parker & Matteson, 2008; Jarvis & Lowe, 2008; Campbell et al., 2010). Interviews with fishers indicate that fish collected from >25 m require more decompression time and venting while at depth, or several venting applications during ascent. Fishers have also mentioned that larger fish exhibit more severe external barotrauma symptoms than smaller fish of the same species, which is similar to findings in studies on deeper-dwelling food fishes (Hannah, Parker & Matteson, 2008; St John & Seyers, 2005). Just as different deeper-dwelling food fish species exhibit different responses to ascent rate (Hannah & Matteson, 2007; Jarvis & Lowe, 2008; Pribyl, 2010), aquarium fish species reportedly react differently to ascent rate and venting. These differences are likely caused by variation in body shape, tissue durability, and swim bladder volume between species. Methods used by fishers reflect these species differences, with practices such as performing venting on more delicate, soft-bodied fish like angelfish (Pomacanthidae) underwater to prevent swim bladder expansion. Examining differences among aquarium fish species of varying sizes and investigating the variety of techniques employed by fishers during collection would provide further insight into the prevalence and effectiveness of aquarium fish barotrauma prevention and mitigation methods.

Histopathology

Histopathology did not detect significant widespread inflammation, organ damage or infection caused by venting. Only one case of a needle wound was found that consisted of localized necrosis and inflammation, with no visible evidence of infection. It is possible that histologic sectioning of tissues missed similar lesions in other fish, but this was minimized by focusing the sampling at the site consistently used by fishers for venting. However, the objective of histopathology in our study was to determine if widespread inflammation or tissue damage was present in fish indicating significant injury, which was not found. If such injuries were present, they would have been detected in multiple sections of the tissues surrounding the venting wound.

Wound healing with no evidence of ongoing necrosis or inflammation, as seen in these fish, indicates that the venting procedure does not pose a significant threat to fish survival post-collection, nor does it cause significant sublethal effects. However, we caution that the fish in our study were held in an aquaculture facility for 21 d without the additional handling and transport stressors they would normally experience in the supply chain, thus potentially promoting recovery from injuries inflicted during collection. It is possible that additional stressors of the supply chain diminish the efficacy of venting in promoting long-term fish survival.

Because aquarium fish exhibited external signs of barotrauma similar to those observed in deeper-dwelling food fishes, we expected internal barotrauma signs to be similar as well. However, we did not detect lesions resulting from barotrauma, even in fish subjected to ascent without decompression. However, externally visible signs of barotrauma did occur. Positively buoyant fish were bloated and had intestinal prolapse at the cloaca. Although not examined in this study, it is likely that organ displacement by the swim bladder occurred in these fish; an internal barotrauma sign observed in deeper-dwelling food fishes (Rogers et al., 2008). Determining if organ displacement occurs, and if venting relieves this issue in aquarium fish would further our understanding of the mechanisms with which venting reduces mortality in fish subjected to ascent without decompression.

Primary stress response

Our results indicate that all collection methods produced elevated cortisol concentrations above the ocean baseline level. Though we did not perform stress treatments on Yellow Tang to determine a cortisol level that corresponds to a stressed state, Soares et al. (2011) did so with a closely related acanthurid (Ctenochaetus striatus). While cortisol concentrations vary between species (Barton & Iwama, 1991), stressed (45–65 ng/ml) and non-stressed (10–25 ng/ml) cortisol concentrations in C. striatus suggest that venting increased stress in fish subjected to ascent without decompression though this was not statistically significant. Despite this increase, we emphasize that venting did mitigate positive buoyancy and ultimately prevented mortality. It appears that venting alone is a short-term stressor, but prevents mortality in fish subjected to ascent without decompression stops.

Future studies should investigate if cortisol levels subside, or remain elevated in the rest of the supply chain. Handling in and transport between export, import, and retail facilities may exacerbate collection-induced stress. Because chronic stress results in immune system suppression (Barton & Iwama, 1991; Barton, 2002), fish experiencing chronic stress are more susceptible to infection, disease, and delayed mortality. Because hobbyists whose aquarium fish die often replace these fish, delayed mortality is a great driver of aquarium fish demand (Tissot et al., 2010). It is likely that stress plays a role in this mortality, and future studies should examine stress as it relates to handling in and transport between each link in the supply chain beyond collection.

Implications for management

While our work adds to scientific knowledge regarding collection practices of aquarium fish in Hawaii, it is also relevant to the global trade. Yellow Tang and other surgeonfish (family Acanthuridae), are one of the most common families targeted globally in the live aquarium trade (Rhyne et al., 2012). Our results also improve our understanding of the effects of venting. Previous studies show conflicting results regarding the effects of venting on fish mortality (Gotshall, 1964; Keniry et al., 1996; Nguyen et al., 2009; Wilde, 2009). Our results indicate that when performed properly, venting does not cause mortality or inflict significant sublethal injuries, though we caution that our inference is limited to a single species.

Though animal rights groups in Hawaii criticize venting, we did not find that it caused mortality or sublethal injuries in Yellow Tang. Banning venting may increase mortality rates if fishers implemented ascent without decompression. While opponents of venting have suggested that slow decompression be used instead, the time required to naturally decompress fish could be economically prohibitive. It is also worth pointing out the dichotomy of venting in the recreational fisheries versus the aquarium trade. Recreational fishers are often encouraged to vent fish before returning them to the water (see Kerr, 2001; Stevely et al., 2011; Theberge & Parker, 2005), while venting remains controversial in the aquarium trade.

In conclusion, we determined that the methods commonly used by aquarium fishers in Hawaii do not cause mortality in Yellow Tang. However, all collection methods produced elevated cortisol concentrations in fish, and this warrants more investigation. Further handling in and transport between links in the supply chain could cause chronically elevated cortisol concentrations in fish, exacerbating stress and minor injuries inflicted during collection.

We thank fishers in Kona, Hawaii, especially Tyron Terrazzono, Paul Masterjohn, and Scott Brien for their time, cooperation, and support. Thanks to Todd Stevenson for project guidance; Syd Kraul for use of his aquaculture facility; Meghan Dailer for her hospitality and encouragement; Tony Spitzack, Cori Kane, Molly Bøgeberg, and Evan Morris for project assistance; Dr. Jim Beets and Caitlin Kryss of The University of Hawaii at Hilo’s Marine Science Department, Dr. Bill Walsh, Laura Livnat, and Kara Osada of the Hawaii Department of Aquatic Resources for logistical support and project guidance; Dr. Bob Jordan and the Kona Veterinary Service for supplies; Ian McComas for centrifuge use, guidance, and his time; Dr. Carl Schreck and Julia Unrein at the OSU Department of Wildlife and Fisheries Laboratory for cortisol analysis and guidance with sampling protocol; Dr. Cheryl Schultz and many others for manuscript edits and suggestions. This work was performed under Washington State University Institutional Animal Care and Use Committee protocol #04151-004.

Additional Information and Declarations

Competing Interests

Author Contributions

Animal Ethics

Data Deposition

Tim Miller-Morgan is an employee of Hatfield Marine Science Center.

Emily S. Munday conceived and designed the experiments, performed the experiments, analyzed the data, contributed reagents/materials/analysis tools, wrote the paper, prepared figures and/or tables, reviewed drafts of the paper.

Brian N. Tissot and Tim Miller-Morgan conceived and designed the experiments, contributed reagents/materials/analysis tools, reviewed drafts of the paper.

Jerry R. Heidel conceived and designed the experiments, performed the experiments, contributed reagents/materials/analysis tools, prepared figures and/or tables, reviewed drafts of the paper.

The following information was supplied relating to ethical approvals (i.e., approving body and any reference numbers):

This work was performed under WSU IACUC protocol #04151-004.

The following information was supplied regarding the deposition of related data:

The National Oceanography Dataset: http://data.nodc.noaa.gov/cgi-bin/iso?id=gov.noaa.nodc:0125562

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
