# Peer review of "The effects of venting and decompression on Yellow Tang (Zebrasoma flavescens) in the marine ornamental aquarium fish trade"

_PeerJ, doi:10.7717/peerj.756_

## Round 0.1 · original submission · Minor Revisions

I asked reviewers with widely differing experience including a fish vet, a public aquarist, and an academic researcher, and all three are enthusiastic about this study and it's value to the field and the management of marine fishes in Hawaii. Each also has a number of suggestions to improve the manuscript. Although the list of suggested revisions is fairly long, I believe that they are relatively straightforward to address. For example, two of the referees reject the notion that collectors cannot use the best possible practice which reduces harm and emotional reactions simply because of economics. I expect that many readers will agree with this sentiment and suggest that the authors would be better sticking to the facts of the science for thier study, which is a valuable contribution regardless of the economic and political debate surrounding the fishery itself. Along these same lines, one question not addressed in the reviews is whether the authors believe that fishers behavior differently when watched? Some might argue that they would be more careful for this study than when working on their own, and it might be good to have a statement to address your thoughts on this given referees questions about using a single fisher for this study. I believe that referees have given you constructive feedback to aid in improving your manuscript, and look forward to your response to referee comments so that I can decide whether or not the referees need to see the manuscript again.

·

Basic reporting

No comment

Experimental design

Experimental design is sound for the study project. No comment

Validity of the findings

Findings are valid. No comment

Additional comments

Throughout the manuscript (beginning with line 132), West Hawaii is used as the descriptive location. Most readers will not know where West Hawaii is. Change to Hawaii or Kona Coast of Hawaii.

In methods/results sections, initial diagnostics were not performed on any fish (live exam or necropsies of dying/sacrificed animals). Examination of biospies to rule out any parasitic infection would have added credibility to paper overall. Also, water chemistry parameters were not measured in collection tanks or 21 day holding areas. Even though these are flow through systems, it would have been good to have those numbers as a rule-out to any further stressors that could influence cortisol levels or mortalities down the transportation process. Please consider adding a few sentences explaining if these diagnostics were performed and/or the reasons they were not included in the study.

Specific editorial comments:
Title: Change "ornamental aquarium fish trade" to "marine ornamental fish trade"
Line 25. References are 9+ old. Is there nothing more recent published on the number of marine ornamentals taken for the fish trade?
Lines 29-30. "sequentially transporting them from collection site to export facility to aquarium fish retail store to the hobbyist aquarium."
Line 34. "destructive fishing practices" is too vague. destructive to who, what? Reefs? Fish, Environment?
Lines 37-38. References are outdated, with most recent from 2005. Recent publications?
Line 50. Here, we examine the first step in the supply chain:"
Lines 52-53. Delete "and ours is the first study to address this problem"
Line 55. Delete "because"
Line 56. "the surface; the water pressure decreases resulting in an increase...."
Lines 64-66. Barotrauma effects of mesopelagic food fishes include protrusion of the esophagus from the mouth (references), swim bladder rupture, internal bleeding...."
Lines 87-88. gases to be removed from the swim bladder. Fishers implement one,...."
Lines 91-93. Delete last sentence, "While the effects of venting and decompression on aquarium fish...fairly well studied in mesopelagic food fishes."
Line 94. "In mesopelagic fishes, decompression takes a long time...."
Line 96. Decompression is a time consuming process. In order to prevent barotrauma,...."
Lines 97-102. "...for fish to naturally remove the expanding gases in the swim bladder. Decompression can be prohibitively time-consuming for fishers to implement even for shallow-dwelling reef fish. Fishers would rather...."
Lines 103-104. "Research on mesopelagic food fishes disagree that venting reduces fish mortality. This is largely an artifact of the differences in species and depths...."
Lines 116-117. "While these methods of barotrauma prevention and mitigation affect fish health and mortality, these practices are controversial among the animal rights community."
Line 119. "(see Lauer 2011; Talbot 2012...."
Line 122. Delete "in"
Line 132. Delete "West". Also see general comments above.
Lines 134 -137. Switch last two sentences around. "In addition, Acanthuridae, the family encompassing...Therefore, understanding how collection practices...and survival is especially relevant to the aquarium industry."
Line 138. Write out "WSU IACUC", followed by acronyms in parentheses.
Line 141. "observation at an aquaculture facility in Kailua-Kona. Fish that died were examined...."
Line 152. " aquarium fish performed the fish collections."
Line 163. "brought up 3 m every 15 min at 10 m (2 atm) depth, these...."
Line 168. "less than three seconds by the fisher...."
Line 175. "Energy Laboratory Hawaii Authority (NELHA) in Kailua-Kona, Hawaii provided natural surface seawater...."
Line 178. "...from the reef to a retailer or hobbyist for this particular chain."
Line 196. Include company in parentheses after "MS-222".
Lines 272-284. This is a summary of the results already stated. Reduce paragraph to few synopsis statements or begin with Mortality section.
Line 290. "treatment group to where there was mortality"
Lines 407-409. Reviewer did not see "Capitini, et.al" referenced in the manuscript.
Lines 492-493. Reviewer did not see "Stevenson and Tissot 2013" referenced in the manuscript.
Lines 501-502. Reviewer did not see Tissot 2005 referenced in the manuscript.

Reviewer 2 ·

Basic reporting

No Comments

Experimental design

I have a few areas of concern.

1. The authors do not state the identity of the fish collector. Perhaps not a must, but #2 is a must.
2. The authors do not state the level of experience of this fisher.
2a. You used only one fisher? That is what this reviewer takes from the section. I have a bit of an issue with only having an n of 1 for such an important part of the study. While this is ok, it is not ideal. The authors need to qualify this in their discussion. There are studies that demonstrate that fishing methods effects mortality in crabs under going declawing. One would think that fishers techniques are a key factor in this study.
3. The authors do not state the date in which the study was conducted in the methods section.


A suggestion:
1. The authors area bit chatty. i.e. the fish feeding section. Please review your methods section. Try and shorten or clean up the section a bit.

Validity of the findings

I find the data compelling in that morality is overall low during the study. Venting and decompression prevent mortality and these findings are important. The authors qualify the data and findings in terms of the duration of the study. The authors do use a holding system that is quite different than that of the average aquarium collector, wholesale facility. The authors also likely hold the fish a much larger period of time than do collectors. I would think that shipping fish with in a few days of collection might be a factor with fish health.

Additional comments

Please include more detailed figures of the histology. Include histology of Day 0 and Day 21 for all treatments. There is no compelling reason not to have images of all treatments and days where histology was taken. This provides a higher level of data reporting.

Lines 36-38. These papers do not all conclude 90% morality. Please clean this up and be more accurate here. I have grown sick of inaccurate accountings revolving trade numbers. You can provide the entire range, or provide the mortality of each study.

It is highly unlikely that the aquarium trade consumes 30 million fish. What are these numbers based on?

Please provide some citations on the hormone levels of fish under stress. There are numerous studies. I'd suggest that the authors can provide a clearer discussion of how stress works in fish. For example, fish acclimate to stress and might show higher levels of hormones than wild fish but these might or might not be stressful levels.


The following statement is absurd. "While opponents of venting have suggested that slow decompression be used instead, the time required to properly decompress these fish is economically prohibitive and impractical for fishers to implement."

Please explain how fishers that are decompressing fish, fishing a population which is the only support of a species in the world, would be economically shut out? If fisherman were required to take their time bring fish up to the surface would price not increase if supply was restricted or if fisherman demanded more for their time? The key would be universal standard. Your statement here has no grounding in the economics of the fish trade. As supply increases so decreases price. I reject the authors statements that fisher would be economically impacted here.

·

Basic reporting

This paper is in reference to an experimental study examining depth of collection, decompression stop, and venting, and how that compares to stress and surivial. They discovered that venting did not increase stress or mortality, and appeared to be a rational management measure.
Overall, this study is well conducted (controlled and replicated). The writing can be tightened in some areas, and likewise, some of the statistical analyses need to be better defined. These will be pointed out below.
Paragraph 2 starting at line 34L is primarily about “possible” mortality. This should be referred to as “unknown levels” – possibility infers causality without proof. If the costs through the supply chain were tracked, significant cost increases would infer mortality at that point. Thus the comments at L42 about economics, could be used to assess where mortality could occur.
L51 – depth needs to be identified.
Paragraph at L64 needs to be converted to a table, and linked to the prior paragraph.
Paragraph at L71 – should be combined with impacts – is this internal sublethal signs, and if so prior paragraph needs to be identified as external signs (currently just signs, although gas emboli in the rete mirabile are not external signs). These paragraphs need to be tightened.
The L71 paragraph also mentions behavioral impairment and infections without further clarification. These need to be better addressed.
L92 – the “has not been documented” statement is confusion. It has been documented (L90), not evaluated (said in L91)m but addressed in food fish (L93). Clarify this sentence.
L95 – “removed from” – prepositional phase
L97 – “naturally remove the expanding swim bladder”?
L98 – “prohibitively time consuming “– how much time? Divers don’t have to stay with fish during decompression.
L108 – “allow for specific conclusions about “ – not sure what these means.

Experimental design

L138 – IACUC protocol should be part of the acknowledgements.
L140 “difference collection methods” – not sure what this means.
L141 – more detailed methods at L145 should occur after “facility in West Hawaii.”
L145 - “three decompression treatments” need to be identified
L153 – N=40 compares to L147 where n=20. Which is correct?
L155- “fish were vented-treatment and half were not” – awkward – rewrite.

Validity of the findings

L256 – need to state results of interaction first because this determined if you need to look at simple effects of main effects.
L258 “all P < 0.05” need to state test. If this is multiple comparisons against a baseline control, then the P value need to be corrected (p/n) for the multiple comparisons.
L259 – df 2,10 is not correct. 3 decompression treatments and 2 venting for 6 treatments with 3 replicates means 18 total observations. There are hence 17 full degrees of freedom (n-1). If venting has 1 df, decompression has 2, and the interaction has 2 (5 total for the model), then 17-5 is 12. Hence the df should be 2,12. This statistic needs to be readdressed.
L262 – stats need to be compared to figure 3 – the numbers provided do not look like what is in the figure. Why is no decompression stops different than many 15 min stops, but not 45 mins? 58.8 different than 35.5 but not 35.2?
L264 – a simple effect is not relevant given that there is not interaction and only main effects are important.
L353 “venting is a short-term stressor” – but not with decompression

Additional comments

If management is going to be discussed, then the rules governing the recreational fishery should be mentioned. It is this reviewers understanding that the rec fishing requires venting, so it appears odd that one fishery would demand it, yet the other consider regulating against it. This dichotomy should be pointed out, particularly given the greater catch by the rec fishery.

---

## Round 0.2 · accepted · Accept

Thank you for your careful and detailed response to the referee comments. You have made changes in response to all the substantive comments and explained your disagreement with a few that I feel are acceptable differences of opinion. Thus, I am happy to accept your manuscript and move it forward to the production staff.